# D3: Data Diversity Design for Systematic Generalization in Visual Question Answering

**Amir Rahimi**                                                                    *arahimi@mit.edu*
*Massachusetts Institute of Technology (MIT)*

**Vanessa D'Amario**                                                               *vanessad@mit.edu*
*MIT / Fujitsu Research of America, Inc.*

**Moyuru Yamada**                                                          *yamada.moyuru@fujitsu.com*
*Fujitsu Limited*

**Kentaro Takemoto**                                                         *k.takemoto@fujitsu.com*
*Fujitsu Limited*

**Tomotake Sasaki**                                                       *tomotake.sasaki@fujitsu.com*
*Fujitsu Limited*

**Xavier Boix**                                                                 *xboix@fujitsu.com*
*MIT / Fujitsu Research of America, Inc.*

**Reviewed on OpenReview:** *https://openreview.net/forum?id=ZAin13msOp*

## Abstract

Systematic generalization is a crucial aspect of intelligence, which refers to the ability to generalize to novel tasks by combining known subtasks and concepts. One critical factor that has been shown to influence systematic generalization is the diversity of training data. However, diversity can be defined in various ways, as data have many factors of variation. A more granular understanding of how different aspects of data diversity affect systematic generalization is lacking. We present new evidence in the problem of Visual Question Answering (VQA) that reveals that the diversity of simple tasks (i.e. tasks formed by a few subtasks and concepts) plays a key role in achieving systematic generalization. This implies that it may not be essential to gather a large and varied number of complex tasks, which could be costly to obtain. We demonstrate that this result is independent of the similarity between the training and testing data and applies to well-known families of neural network architectures for VQA (i.e. monolithic architectures and neural module networks). Additionally, we observe that neural module networks leverage all forms of data diversity we evaluated, while monolithic architectures require more extensive amounts of data to do so. These findings provide a first step towards understanding the interactions between data diversity design, neural network architectures, and systematic generalization capabilities.

## 1 Introduction

Systematic generalization is a crucial aspect of human intelligence Lake et al. (2019), allowing us to solve novel tasks by combining knowledge gained from previously seen, related subtasks Bahdanau et al. (2019b); Lake & Baroni (2018); Ruis et al. (2020). Deep learning methods have traditionally struggled to achieve systematic generalization due to their limited ability to generalize beyond the patterns and examples present in the training data, particularly in the presence of complex compositional and structural biases Bergen et al. (2021); Gontier et al. (2020); Hudson & Manning (2018); Tsarkov et al. (2021). In response, researchers have

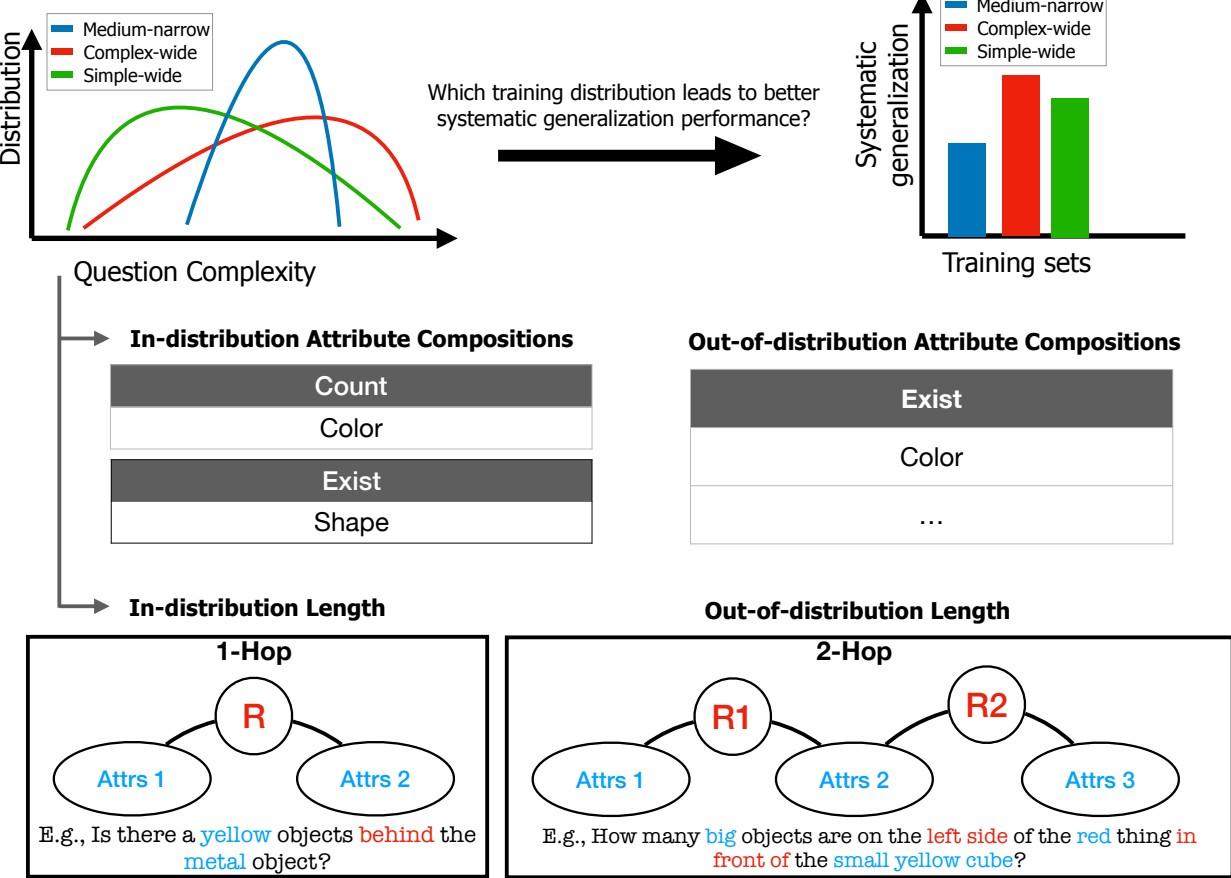

Figure 1: Data diversity and its impact on systematic generalization in VQA. **(top)** Impact of question complexity distribution on systematic generalization. In our study, question complexity is varied in two aspects: **(middle)** Attribute composition. In this example, the training set (in-distribution) contains `Count` and `Exist` question with different compositions (left) while `Count` or `Exist` questions have novel combinations of attributes in the test set (out-of-distribution). **(bottom)** Length. The in-distribution question has shorter length (different syntactic structure) compared to the out-of-distribution question.

recently developed various architectures and training regimes to improve systematic generalization Bahdanau et al. (2019a;b); D'Amario et al. (2021); Kamata et al. (2023); Yamada et al. (2023). Yet, studies prove that achieving systematic generalization remains very difficult Ruis & Lake (2022).

Recent studies have highlighted the critical role of diversity in the training data, as it has been shown to impact systematic generalization performance significantly Madan et al. (2022); Ruis & Lake (2022). Madan et al. Madan et al. (2022) demonstrated that increasing training data diversity substantially improves generalization to out-of-distribution (OOD) category-orientation combinations. Similarly, Ruis and Lake Ruis & Lake (2022) showed that augmentation of training data and modularity increases systematic generalization performance substantially in a natural language compositional generalization task. However, there is still a lack of study on which specific aspects of data diversity are responsible for enhancing systematic generalization and under what conditions and how they can be applied effectively. This calls for further investigation to unravel the relationships between different types of data diversity and their impact on systemic generalization.

Figure 1 illustrates a motivating example where given a budget of $N$ training questions and a test set in Visual Question Answering (VQA) Agrawal et al. (2018); Antol et al. (2015); Kervadec et al. (2021), training on different question complexity distributions results in different systematic generalization performances. To study the impact of question complexity on systematic generalization, we consider two factors that influence question complexity: (i) *Attribute composition* that focuses on specific combinations of attributes and question types during training. For instance, a model is trained on questions that involve only certain combinations,

such as `Count` questions with `Color` attributes and `Exist` questions with `Shape` attributes. However, during testing, the model needs to generalize to novel combinations of questions and attributes, such as `Exist` questions with `Color` attributes. (ii) *Length* that refers to the number of reasoning steps required to answer the question[1]. In our study for systematic generalization, we require a model to generalize to novel question lengths. We measure question length by the number of spatial relations it includes. Spatial relations refer to relations such as `"behind"`, `"left"`, `"right"`, and `"in front of"`, between entities in an image. The hop notation, such as `0-Hop`, `1-Hop`, and `2-Hop`, is used to categorize questions based on the number of spatial relations they involve. The hop count indicates the number of reasoning steps (analogous to the number of sub-tasks) required to answer a question. Higher hop counts imply more advanced spatial reasoning and compositional understanding. An example for length generalization is illustrated in the bottom part of Figure 1.

In this paper, we investigate various factors in designing training data that contribute to improving systematic generalization and explore the conditions under which these factors can be effectively applied. By creating datasets in controlled settings in VQA, we analyze the relationship between different factors in training question complexity (e.g., length and compositions of object attributes) and systematic generalization performance. Our empirical analysis reveals that simple diverse questions, i.e., questions that are simple (require less reasoning steps), but cover a diverse range of attributes, are effective for systematic generalization. This can be valuable for practitioners and dataset engineers since collecting simple questions and obtaining their associated answers is more cost-effective than curating a large number of complex questions with many attributes and spatial relations. Moreover, simple questions facilitate the development of models that are less susceptible to overfitting. This is because obtaining a diverse representation of attributes through unbiased sampling is considerably more manageable with simple questions as opposed to complex ones. Our finding is in line with recent research Sorscher et al. (2022) that highlighted the inefficiency of solely relying on the neural scaling law Gordon et al. (2021); Hestness et al. (2017); Hoffmann et al. (2022); Rosenfeld et al. (2020); Zhai et al. (2022) to enhance model performance through data scaling. Instead, it has been shown that designing datasets strategically based on difficulty and dataset size can lead to greater cost efficiency without compromising performance.

## 2 Data Diversity Benchmark for Systematic Generalization in VQA

In this section, we describe the datasets we generate using the CLEVR Johnson et al. (2017a) repository. We define biased training questions where the biases are formed by limiting the attribute compositions and question lengths in the training sets while we test on out-of-distribution questions. Our objective is to gain insights into the models' capabilities and limitations in different aspects of systematic generalization.

We propose our Data Diversity Design, referred to as D3, for each task which involves incorporating diverse set of questions to enhance systematic generalization. By doing so, our methodology helps mitigate the impact of biases in the training data, thereby allowing VQA models to generalize effectively to previously unseen question attribute compositions and lengths.

We begin by introducing biases in simple questions about the composition or comparison of a set of objects using two attributes, focusing on compositional generalization. Next, we incorporate more complex questions regarding the spatial relations between multiple sets of objects to incorporate length generalization and explore various aspects of systematic generalization. Finally, we explore the impact of question complexity distribution as an additional aspect of diversity on systematic generalization when full diversity is present.

### 2.1 Attribute Composition Generalization

Here, we provide detailed description of two biased training sets, sets used for applying D3 (also called D3 sets), and systematic generalization test sets for evaluating attribute composition generalization.

**Base training set for compositions of two attributes.** This dataset comprises simple elemental questions involving a composition of two attributes as the biased in-distribution (InD) set for training. The images have been generated in a way that three to five objects are present in each image, thereby resulting in a

---

[1]Not to be confused with the number of words in a question in this context

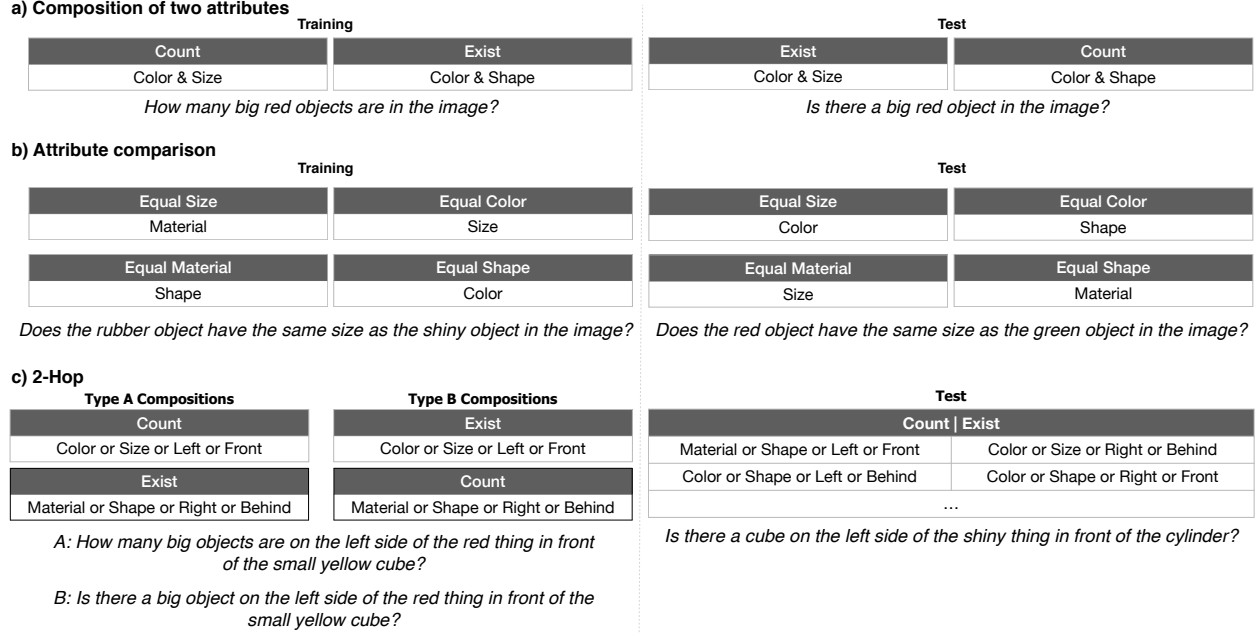

Figure 2: Specification of different biases for train and test datasets. The gray boxes display the question type, while the valid attributes corresponding to each question type are shown in white boxes below. A valid example question for each dataset is shown below the boxes.

simplified dataset. We have used the original `0-Hop` template from the CLEVR dataset to generate InD and OOD questions for this dataset. Additionally, we have imposed constraints on the types of questions to specify the biased train and test splits.

We have limited the pairs of attributes and question types appearing together in this dataset to only `Count` and `Exist` question types, as shown in Figure 2a. Specifically, `Count` questions are only associated with `Color` and `Size` related attributes, while `Exist` questions are employed only when `Color` and `Shape` attributes are present in the questions. For instance, the question *"How many big red objects are in the image?"* is a valid InD question for this dataset as it contains two attributes relating to `Color` and `Size` for the `Count` question type. However, the question *"How many rubber objects are there?"* will never appear during training in this dataset.

**Test set for composition of two attributes.** The systematic generalization test set for this dataset contains questions where the attributes associated with `Count` and `Exist` questions are swapped with respect to InD questions. For instance, in the OOD test set, `Color` and `Size` attributes are used in `Exist` type questions. By designing this test set, we can evaluate if the neural networks have learned to identify and understand the elemental attributes and their relationships to different question types.

**Base training set for attribute comparison.** The dataset for attribute comparison includes four question types: `equal_size`, `equal_shape`, `equal_material,` and `equal_color`. Similar to the composition of two attributes dataset, the images are generated such that three to five objects are present in each image. For each question type, a specific attribute of the same type between two different objects is compared. Different attribute and question type combinations for the training set of this dataset is shown in Figure 2b. As an example, `equal_size` type questions always filter `Materials` of two objects in an input image and compare if they have the same `Size`. In this case, a question like *"Does the rubber object have the same size as the shiny object in the image?"* is considered a valid question. Note that for simplicity, there is no combination of attributes of objects in these type of questions.

**Test set for attribute comparison.** The OOD test set for this dataset is constructed in a way that the comparison questions appear with different attributes from the training set as shown in Figure 2b. For example, the `equal_size` questions are about the `Color` attributes of the objects in the test set.

**Applying D3.** To apply D3, we utilize `0-Hop` questions that have no spatial relations with a constraint that questions from this set contain only a single attribute. Our diversified training sets after D3 contain 70% of the original biased questions and 30% of the questions from the respective D3 set. While we have observed promising results with the 70% and 30% configuration, we have also explored other proportions, which will be detailed in Appendix B. Our goal is to investigate the impact of incorporating such simple questions on systematic generalization. We use questions about `Color, Size,` and `Shape` attributes for both `Exist` and `Count` questions. As an example, the question *"Is there a shiny object?"* is a valid question for this set.

## 2.2 Incorporating Length Generalization

To analyze the interplay between different aspects of diversity for systematic generalization, we introduce a set of datasets with varying compositions of attributes and question lengths. The biased training set is based on the `2-Hop` template, which includes questions with two spatial relations. This is a more complex setting where spatial relations between objects and attributes come into play and allows us to study how the complexity of the questions and the diversity of attributes can affect the model's ability to generalize systematically. By exploring different combinations of attributes and question lengths, we can gain insights into which factors contribute most to achieving high levels of systematic generalization in VQA models. The images from the original CoGenT A and CoGenT B splits of CLEVR are used for the training and test sets, respectively. We only consider `Count` and `Exist` question types for these sets. Different variations of the sets are illustrated in Figure 2c. Next, we begin by introducing the biased training set, OOD test sets, and how we apply D3.

**Base training set for two spatial relations between multiple objects (2-Hop).** The biased training set for this dataset is named `2-Hop A`, in which only `Color, Size, Left,` and `Front` attributes are allowed for `Count` type questions, and `Material, Shape, Right,` and `Behind` attributes are used for `Exist` questions.

**Test sets for 2-Hop.** We introduce four different test sets to evaluate attribute composition generalization, length generalization, and their combinations using the `2-Hop` dataset:
- *0-Hop:* The `0-Hop` test set is composed of questions regarding a single attribute. It contains all possible combinations of attributes and question types (i.e., `Count` and `Exist` questions) . We employ this test set to assess the models' ability to generalize to questions with shorter lengths.
- *2-Hop OOD:* This set contains the same set of combinations for both `Exist` and `Count` question types. The attribute and spatial relation combinations are carefully chosen such that the possible combinations have an edit distance of two with attribute combinations in `2-Hop A`. For example, `Material, Shape, Left,` and `Front` can appear with both `Count` and `Exist` questions. This set is solely used for evaluating attribute composition generalization on the more complex `2-Hop` questions than the previously introduced attribute composition test sets.
- *3-Hop A:* This test set comprises questions with three spatial relations and uses only type `A` attribute compositions. The sole purpose of this test set is to evaluate the network's ability to generalize to longer questions.
- *3-Hop OOD:* This test set is similar to `2-Hop OOD` in terms of attribute and question combinations. However, the questions contain three spatial relations (`3-Hop`) , making it a more challenging set that tests both OOD length and OOD attribute composition generalization.

**Applying D3.** The sets for applying D3 consist of various combinations of attributes and spatial relations, enabling us to systematically analyze the model's ability to generalize to different levels of complexity. As before, we replace 30% of the base training questions with questions from one of the sets below to create a new diversified training set:

- *1-Hop Full:* This set includes questions that have only one spatial relations between two objects, hence called `1-Hop Full`. There is no combinations of attributes for a single set of objects in this D3 set. We do not impose any constraint on the combinations of questions and attribute types for the `1-Hop Full` set. This set is analogous to the diverse simple D3 set of questions that we used for attribute comparison and composition sets for questions with spatial relations.
- *1-Hop A:* The `1-Hop A` set contains the same combinations of questions and attributes as the `2-Hop A` training set, but with only one spatial relation. This allows us to study the model's ability to generalize to

Table 1: *Accuracy and standard deviations comparing training on the biased datasets and training after D3 with* `0-Hop`*. 30% of the training questions from biased datasets are replaced with* `0-Hop` *questions for D3.*

| Training dataset | FiLM | MAC | VectorNMN GT | VectorNMNSepStem GT | VectorNMN |
|---|---|---|---|---|---|
| Two attributes | $33.57 \pm 0.03$ | $46.56 \pm 0.02$ | $45.39 \pm 0.01$ | $44.10 \pm 0.01$ | $42.80 \pm 0.01$ |
| + D3 (`0-Hop`) | $64.21 \pm 0.03$ | $71.09 \pm 0.04$ | $66.72 \pm 0.01$ | $64.88 \pm 0.01$ | $60.15 \pm 0.01$ |
| Comparisons | $49.28 \pm 0.01$ | $53.31 \pm 0.01$ | $66.27 \pm 0.04$ | $67.97 \pm 0.04$ | $58.43 \pm 0.04$ |
| + D3 (`0-Hop`) | $56.29 \pm 0.03$ | $56.58 \pm 0.03$ | $83.29 \pm 0.02$ | $76.16 \pm 0.04$ | $73.32 \pm 0.02$ |

longer questions and analyze its similarity to the test distribution.

- *1-Hop B:* This set is similar to `1-Hop A` except that the attribute and spatial relations of `Count` and `Exist` are swapped from the ones in `1-Hop A`. Specifically, in `1-HopB`, `Exist` questions can only be paired with `Color, Size, Left`, and `Front` attributes. This set serves the same purpose as `1-Hop A`, allowing us to analyze the model's similarity to the test distribution.

- *3-Hop Full:* This set comprises `3-Hop` questions with no restrictions on the combinations of question types and attributes. The main purpose of this set is to evaluate the model's ability to generalize to questions with shorter lengths than the ones seen during training.

## 2.3 Question Complexity Distribution

In order to explore the impact of question complexity distribution, we introduce the datasets with full diversity. These datasets allow us to examine the systematic generalization performance by sampling questions from various question types in different proportions, while maintaining a fixed number of questions.

**Training sets.** We sample different fractions of training questions from `0-Hop A`, `1-Hop Full`, and `2-Hop A` training sets. `1-Hop Full` and `2-Hop A` are the same sets as the ones we defined in Section 2.2. The `0-Hop A` set is similar to `2-Hop A` introduced in Section 2.2 and Figure 2c except that it does not contain spatial relations. In other words, the `Count` questions appear with `Color or Size` attributes, and `Exist` questions appear with `Material or Shape` attributes. Only a single attribute can be used in the questions of the `0-Hop A` set. A valid question for this set would be *"Is there a matte object in the image?"*

**Test sets for question complexity distribution.** The test sets for these wide datasets are identical to the test sets we defined for the `2-Hop` dataset in Section 2.2.

**Applying D3.** For simplicity, we sample different fractions from each training set using only multiples of 1/6 as a fraction of questions sampled from each training set. We generated 13 and 10 different sets of fractions for 100k and 600k questions, respectively.

## 3 Results

In this section, we provide our results and analysis about the effect of dataset diversity on systematic generalization. Our implementation is based on the original implementation of Bahdanau et al. (2019a)[2]. We used MAC Hudson & Manning (2018), FiLM Perez et al. (2018), Vectorized Neural Module Networks Andreas et al. (2016); Bahdanau et al. (2019a) (VectorNMN), its counterpart with given program generator (VectorNMN GT), and the version introduced in D'Amario et al. (2021) with modular image encoders (VectorNMNSepStem). To save time and computational resources, we excluded FiLM and VectorNMNSepStem from our experiments about question complexity distributions due to their inferior performance compared to MAC and VectorNMN. We conducted hyperparameter search using the attribute comparison and composition datasets and identified a set of effective hyperparameters that yielded consistent results. Further implementation details and information on the hyperparameter search can be found in Appendix A. To generate InD and OOD objects in the images, we rely respectively on the CoGenT condition A and CoGenT condition B splits in CLEVR.

### 3.1 Effects of Introducing Diversity in Training Data on Systematic Generalization

The results for composition of two attributes and attribute comparison datasets (Section 2.1) are shown in Table 1. We also present the full results for the `2-Hop` datasets (Section 2.2) in the form of a heatmap

---
[2]https://github.com/rizar/CLOSURE

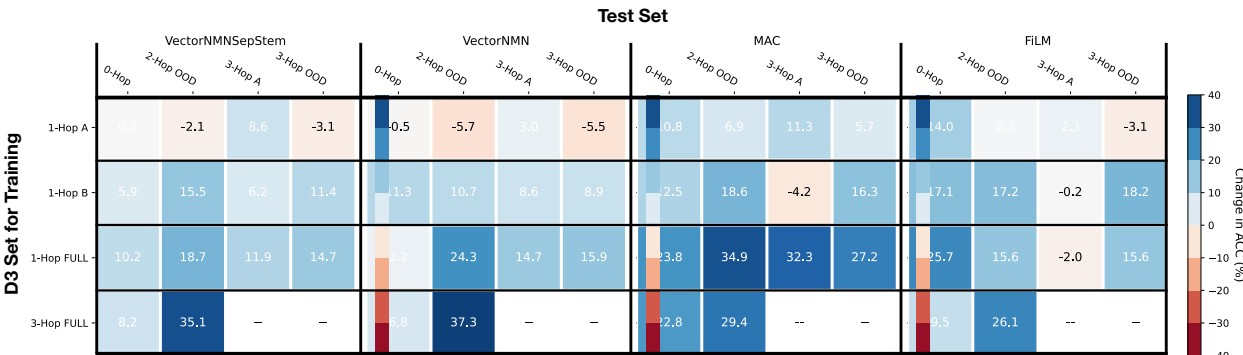

Figure 3: Change in accuracy after replacing 30% of base training questions (`2-Hop A`) with different type of questions as D3 to have more diversity. In most cases diversity helps systematic generalization significantly.

displayed in Figure 3. The heatmap shows the change in accuracy after applying the respective D3 compared to training on the base `2-Hop A` dataset. The results on the base `2-Hop A` dataset are shown in Table 7 in Appendix E. Additinoally, we present the results for a transformer model in Table 5 in Appendix E. As expected, our observations indicate that diversity plays a crucial role in enhancing systematic generalization across various scenarios and D3 with maximum diversity leads to better systematic generalization. However, we note that when the diversity aligns more closely with the specific aspect being tested, we observe improved systematic generalization performance. The key findings are detailed in the following:

**D3 with diverse simple questions improves both compositional and length generalization.** The results for compositions of two attributes and attribute comparisons presented in Table 1 show that using diverse simple questions of `0-Hop` type with only a single attribute as D3 can significantly improve attribute composition generalization for both of our simple datasets. To specifically examine the impact of diverse simple questions within the D3 framework, we analyze the results of the `2-Hop OOD` test set, displayed in the second column of Figure 3, for the respective architectures. Notably, we observe noteworthy improvements in systematic generalization performance when considering both the `1-Hop Full` and `1-Hop B` sets, which demonstrate diversity in attribute composition aspect.

We employ `3-Hop A` test set to isolate the impact of data diversity on length generalization. This test set maintains the same attribute composition as the biased training set `2-Hop A`, but with longer questions. Remarkably, we found that introducing diversity through shorter questions with the same attribute compositions, as seen in `1-Hop A`, improves OOD generalization by incorporating more variations in length. Additionally, we observe that including `1-Hop B` slightly improves the results. It shows that although it introduces more diversity in attribute compositions along adding diversity in length, the networks may learn to associate questions with type B attribute compositions with shorter questions, making it more challenging to generalize to longer questions. On the other hand, since `1-Hop Full` contains all forms of compositions and shorter question lengths altogether, the networks learn a better strategy to generalize to longer questions.

To examine the influence of length and attribute composition together, we utilize the `3-Hop OOD` and `0-Hop` test sets. In the case of `3-Hop OOD`, D3 with `1-Hop A` results in a slight decrease in accuracy across all architectures, except for MAC, which demonstrates an improvement of 5.7%. Conversely, D3 with `1-Hop B` leads to a significant increase in accuracy. This highlights the importance of diversity in both length and attribute composition, as `1-Hop B` exhibits such diversity while `1-Hop A` only diversifies in terms of length. Furthermore, in line with previous observations, `1-Hop Full` outperforms the other D3 sets on the `3-Hop OOD` test set, indicating superior generalization capabilities when full compositional diversity is available. We observe that all forms of D3 lead to improvements on the `0-Hop` dataset (with exclusion of VectorNMN when D3 uses 1-Hop A). For instance, incorporating `1-Hop A` D3, which diversifies only the length aspect of the questions compared to the base training of `2-Hop A`, results in noticeable improvements. On the other hand, D3 with `1-Hop B`, which is more diverse in terms of both length and attribute composition, outperforms D3 with `1-Hop A`. The best performance is achieved by `1-Hop Full` as it is closer in length to the test set and contains full attribute composition diversity. Despite `3-Hop Full` having longer question length than

the base training and thus a further distance to the `0-Hop` test set in terms of length, it still exhibits full attribute diversity and yields significant improvements over the baseline. In conclusion, the diversity of both length and attribute compositions proves beneficial for the `0-Hop` test set.

**D3 with maximum attribute composition diversity leads to better results for attribute composition generalization.** Comparing the performance of `1-Hop B` and `1-Hop Full`, we observe a notable difference when it comes to having the same attribute compositions as the base `2-Hop A` training set. Specifically, `1-Hop Full`, which shares the attribute compositions with `2-Hop A`, demonstrates superior performance. The reason is that `1-Hop B` introduces a swap in attribute compositions for `Count` and `Exist` questions compared to `2-Hop A`. While this diversification contributes to increased variability in the dataset, it causes the model to struggle to generalize effectively to new attribute compositions, leading to a slightly lower performance compared to `1-Hop Full`. Conversely, when focusing on D3 with `1-Hop A`, which solely introduces diversity in terms of question length while keeping attribute compositions constant, we observe minimal changes in performance.

**D3 enhances systematic generalization without getting closer to the target distribution.** Since diversity provides a better generalization to OOD questions, one may ask if by applying D3, the training set is getting closer to the target OOD distribution. To answer this question, we provide the following observations:

- *Shorter questions facilitate length generalization on longer questions, and vice versa:* We observe that any form of D3 utilizing `1-Hop` variants of questions leads to improved results on `3-Hop A`, which contains longer questions compared to the ones in the D3 set. Similarly, employing `3-Hop Full` D3, which consists of longer questions than the base `2-Hop OOD` training set, enhances accuracy on the `0-Hop` test set, which comprises shorter questions.

- *D3 with questions that exhibit more diversity tend to yield better results while maintaining the same distance in terms of attribute composition to the OOD test set:* This can be observed when comparing D3 with `1-Hop A` and `1-Hop B`. Both D3 sets have the same attribute compositions, but they differ in their question types (only `Exist` and `Count` questions have been swapped). Consequently, their distance to the `2-Hop OOD` test set remains constant since the `2-Hop OOD` test set encompasses identical attribute compositions for both `Exist` and `Count` questions. However, the resulting set after D3 with `1-Hop B` exhibits more diversity and gains better accuracy.

These observations highlight an important aspect of D3 training: it does not necessarily bring the training data distribution closer to the target test distribution, yet it can still improve systematic generalization.

We also note that it is crucial to consider the alignment between the diversity introduced in the training data and the characteristics of the target test set to obtain optimal performance. For instance, when evaluating the performance on the `0-Hop` test set, we observed that the `3-Hop Full` D3 set, which consists of longer questions compared to the base `2-Hop A` training set, results in worse accuracy compared to the `1-Hop Full` D3 set. This is because the questions with longer length in the `3-Hop Full` set created a greater distance between the training and test distributions in terms of question length, affecting the generalization performance. In contrast, the `1-Hop Full` D3 set, which encompassed a more similar question length distribution to the target test set, exhibited better accuracy on the `0-Hop` test set.

**Qualitative results.** Qualitative results of MAC trained on `1-Hop` variants of D3 on `2-Hop OOD` test set are shown in Figure 4. Consider the top-left question-image pair. The question has the type of `Count` and it contains `Color, Behind, Size,` and `Right` attributes. As we can see in Figure 2, `Right` and `Behind` attributes are always present in `Exist` type of questions in `Type A` datasets. This has fooled the models trained on `2-Hop A` and `+D3(1-Hop A)` in determining that the question has actually `Count` type resulting in wrong answers. On the other hand, models trained on `+D3(1-Hop B)` and `+D3(1-Hop FULL)` are able to correctly answer the question. We computed the number of questions that the base dataset `2-Hop A` answers a number to questions of type `Exist` as well as `Yes/No` answers to `Count` questions. `2-Hop A` makes such mistakes in determining the type of questions in about 48% of the time, while these type of mistakes for `+D3 (1-Hop FULL)` is reduced to 9%. These type of mistakes also happen in some other image-question pairs in Figure 2. As an example, in the top right image-question pair, `2-Hop A` made a mistake in determining the type of question although the number it provided (0) is correct. The bottom right pair shows a failure case for all of the datasets.

**Q:** There is a purple thing behind the large purple thing; what number of tiny blue things are to the right of it? **Answer:** 1

**Preds (2-Hop A, +1-Hop A, +1-Hop B, +1-Hop (Full)):** no yes 0 1

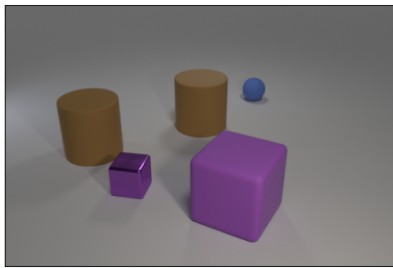

**Q:** Are there any big balls that are in front of the large thing in front of the large cylinder? **Answer:** no

**Preds (2-Hop A, +1-Hop A, +1-Hop B, +1-Hop (Full)):** 0 no no no

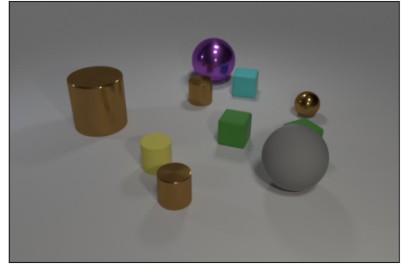

**Q:** There is a brown object that is behind the cyan rubber thing; are there any green metallic objects in front of it? **Answer:** yes

**Preds (2-Hop A, +1-Hop A, +1-Hop B, +1-Hop (Full)):** 2 3 no yes

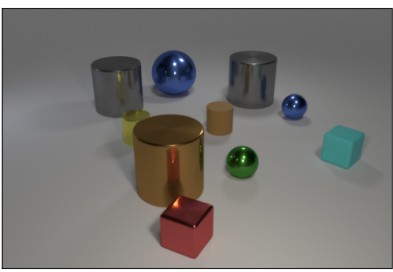

**Q:** How many matte things are right of the big thing that is in front of the big metal object? **Answer:** 2

**Preds (2-Hop A, +1-Hop A, +1-Hop B, +1-Hop (Full)):** 0 0 2 2

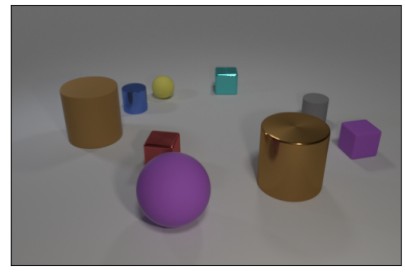

**Q:** Are there any objects behind the cylinder on the left side of the yellow object? **Answer:** yes

**Preds (2-Hop A, +1-Hop A, +1-Hop B, +1-Hop (Full)):** 2 yes yes yes

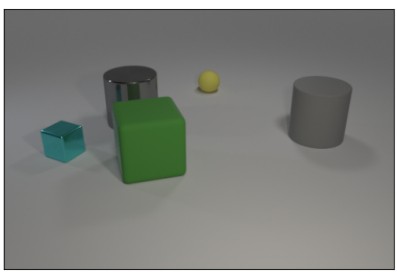

**Q:** What number of yellow rubber objects are in front of the object that is behind the red metal thing? **Answer:** 1

**Preds (2-Hop A, +1-Hop A, +1-Hop B, +1-Hop (Full)):** 0 0 0 1

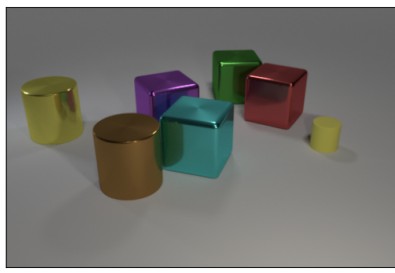

**Q:** Is there a block that is behind the cube to the left of the tiny ball? **Answer:** no

**Preds (2-Hop A, +1-Hop A, +1-Hop B, +1-Hop (Full)):** 1 yes no no

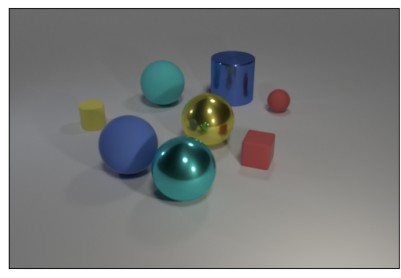

**Q:** There is a object that is to the left of the gray object; how many things are on the right side of it? **Answer:** 3

**Preds (2-Hop A, +1-Hop A, +1-Hop B, +1-Hop (Full)):** no no 0 2

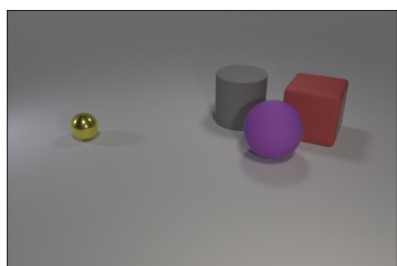

Figure 4: Qualitative results. The questions and ground truth answers are displayed in the top line above each image, while the predictions of MAC using `2-Hop A, D3(+1-Hop A), D3(+1-Hop B), D3(+1-Hop (Full)))` are presented from left to right in the second line above each image.

## 3.2 Additional Datasets and Models

So far, we have performed our experiments on CLEVR dataset using traditional VQA models. The reason for focusing on datasets in controlled settings is that they better ensure that the systematic generalization accuracy is not due to shortcuts in the dataset. It is unclear what biases and shortcuts are present in unconstrained datasets, and hence, unconstrained datasets are limited for gaining an understanding of the systematic generalization capabilities of neural networks. Yet, we recognize the necessity for additional analysis of the contribution of D3 and its application to more unconstrained datasets and state-of-the-art models. In the following we provide additional results on non-synthetic datasets. Furthermore, we provide supplementary results on models based on Large Language Models (LLM) that are significantly larger than the traditional models we have experimented with.

**TallyQA dataset.** TallyQA Acharya et al. (2019) is an open-ended counting dataset for VQA that contains questions generated by humans on real images. We created two splits using the TallyQA training set to experiment with both compositional and length generalization:

1. **TallyQA-Nouns.** Our goal in generating the TallyQA-Nouns dataset was to utilize questions containing a certain number of nouns for training, while incorporating diversity in the number of nouns within the questions. We aimed to test for OoD number of nouns in the questions. To identify nouns, we employed off-the-shelf part-of-speech tagging Honnibal et al. (2020). Before performing part-of-speech tagging, we preprocessed the questions by removing repetitive phrases such as "*In how many of these screenshots...*" or "*in the photo?*" from the beginning or end, as these phrases may introduce additional nouns. We curated two biased datasets: `2Nouns`, comprising about 62,000 questions containing two nouns, and $3^+$`Nouns`, consisting of about 17,000 questions with three or more nouns. Additionally, we included a set of questions with only one noun, referred to as `1Noun`, as the D3 set for the biased datasets. Furthermore, we created an additional split using $3^+$`Nouns` as the D3 set for the biased `2Nouns` dataset. The evaluation was conducted on the full test set of the TallyQA dataset, and the results of the MAC model are presented in Table 2. The results demonstrate that incorporating D3 can enhance performance on real-world, in-the-wild questions across all scenarios. Specifically, we observe a 50% relative improvement for $3^+$`Nouns` datasets. By comparing the performance of the models trained with `2Nouns + D3(1Noun)` and `2Nouns + D3(3`$^+$`Nouns)`, we validate our hypothesis that simpler questions lead to better generalization within this dataset as well.

2. **TallyQA-Cars**. The goal of this split is to assess the combination of length and compositional generalization. Below are the specifications of the splits:
   - `Cars-InD`: All InD questions are derived from the `2Noun` set defined above, with the constraint that questions containing a color word (such as "`white`", "`pink`", or "`black`") are included only if the word "`car(s)`" is also present in the question. This ensures that questions about colors are only included when they are associated with cars. All other `2Noun` questions that do not contain any color or car are included in this set.
   - `Cars-D3`: The D3 set consists of questions that are `1Noun` questions and contain a color word.
   - `Cars-OOD`: The OOD set comprises questions with $3^+$`Nouns` that contain color words but not in conjunction with cars.

   This dataset is designed to evaluate whether D3 enhances the composition of colors with words other than car/cars in longer questions encountered during training. The results of the MAC evaluation are presented in Table 3. In alignment with previous results, we observe that D3 enhances compositional and length generalization on this split.

**GQA Dataset and GPT-2 style model.** We devised the following experiment for length generalization on the GQA dataset Hudson & Manning (2019). We measure length as the number of semantic operations for each question provided in the dataset. We observed that about 47% of the questions in the balanced training set contain three semantic operations. We used these questions as our biased base training set. About 28% of

Table 2: *Accuracy (%) of MAC on TallyQA-Nouns.*

| Dataset | Test Accuracy |
|---|---|
| 2Nouns | 40.67 |
| $3^+$Nouns | 22.88 |
| 2Nouns + D3(1Noun) | 43.63 |
| 2Nouns + D3($3^+$Nouns) | 40.91 |
| $3^+$Nouns + D3(1Noun) | 34.36 |

Table 3: *Accuracy (%) of MAC on TallyQA-Cars.*

| Dataset | Cars-OOD Accuracy |
|---|---|
| Cars-InD | 63.94 |
| + D3 | 69.47 |

the questions have two semantic operations. We call this the set of short questions. The remaining 25% of the questions contain four to nine semantic operations. We call this the set of long questions. We constructed two different D3 sets by replacing 30% of base questions with short or long questions. Our constructed training sets have a total of 450k questions. We trained MAC and a GPT-2 style transformer with an additional fixed vision encoder (Resnet-101) on this dataset and tested on out-of-distribution length questions. Results confirm the previous observations. Both MAC and transformer based models got a boost in accuracy when the D3 set contained shorter questions and tested on unseen long questions as shown in Table 4. In addition, we report the results on GPT-2 Radford et al. (2019) style transformer, trained from scratch, with a fixed vision encoder (pretrained Resnet 101) on our 2-Hop datasets in Table 5. All of the results align well with our findings on the CLEVR dataset and traditional models.

Table 4: *Change in accuracy (%) for length generalization on GQA dataset.*

| Model, Train / Test | Short | Long |
|---|---|---|
| MAC, D3(+short) | - | +1.94 |
| MAC, D3(+long) | +0.2 | - |
| GPT-2, D3(+short) | - | +1.35 |
| GPT-2, D3(+long) | +1.63 | - |

Table 5: *Results of GPT-2 style transformer on 2-Hop datasets.*

| Train set | 0-Hop | 2-Hop OOD | 3-Hop A | 3-Hop OOD |
|---|---|---|---|---|
| 2-Hop A | 12.9 | 23.7 | 45.2 | 20.6 |
| + D3 (1-Hop FULL (30%)) | 29.1 | 59.7 | 45.0 | 44.6 |
| Improvement | +16.2 | +36.0 | −0.2 | +24.0 |

**MiniGPT-v2.** MiniGPT-v2 Chen et al. (2023) is a vision language model that aligns a frozen visual encoder, such as ViT Dosovitskiy et al. (2021), with a Large Language Model (LLM), such as Llama-2 Touvron et al. (2023), using a linear projection layer. We utilized Llama-2 Chat 7b as the language model in all the experiments. Initially, we evaluated the off-the-shelf trained model checkpoint of MiniGPT-v2 (after stage 3) on our 2-Hop OOD dataset. Interestingly, MiniGPT-v2 only achieved an accuracy of 34.9%, despite being trained on an extensive number of VQA datasets. Subsequently, we followed their exact instructions to fine-tune MiniGPT-v2 on our 2-Hop datasets. The changes in accuracy results after applying various D3 sets are illustrated in Figure 5. The base results of training on 2-Hop A are also presented in Table 7 in Appendix E. These results also align with findings from other models. D3 enhances systematic generalization, and diversity plays an important role for MiniGPT-v2. Once again, we observe that in most cases, D3 improves the results, with D3 combined with 1-Hop FULL achieving the best performance.

### 3.3 Effect of Question Complexity Distribution on Systematic Generalization

We have shown that using as much diversity as possible is beneficial for systematic generalization. However, the impact of question complexity distribution, specifically the question sampling strategy, remains unknown. The

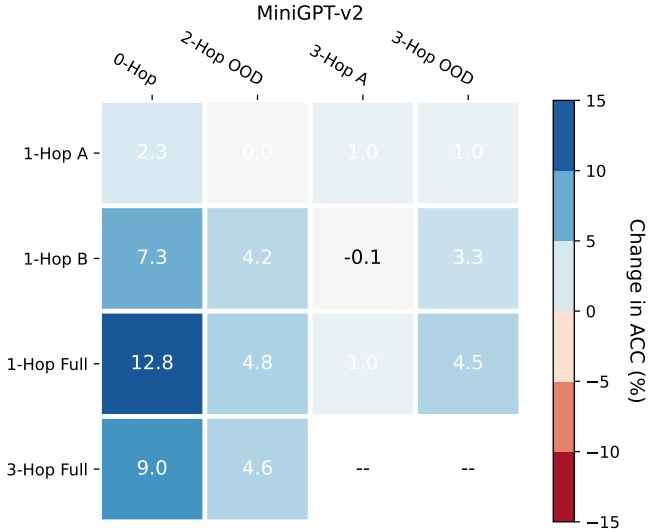

Figure 5: Change in accuracy after applying D3 on `2-Hop A` dataset using the MiniGPT-v2 model.

conventional method of uniformly sampling questions from different levels of complexity may be suboptimal for systematic generalization. In the following set of experiments, our goal is to identify the impact of question complexity distribution, the number of training questions, and the behaviour of different neural network architectures in low and large data regimes on systematic generalization.

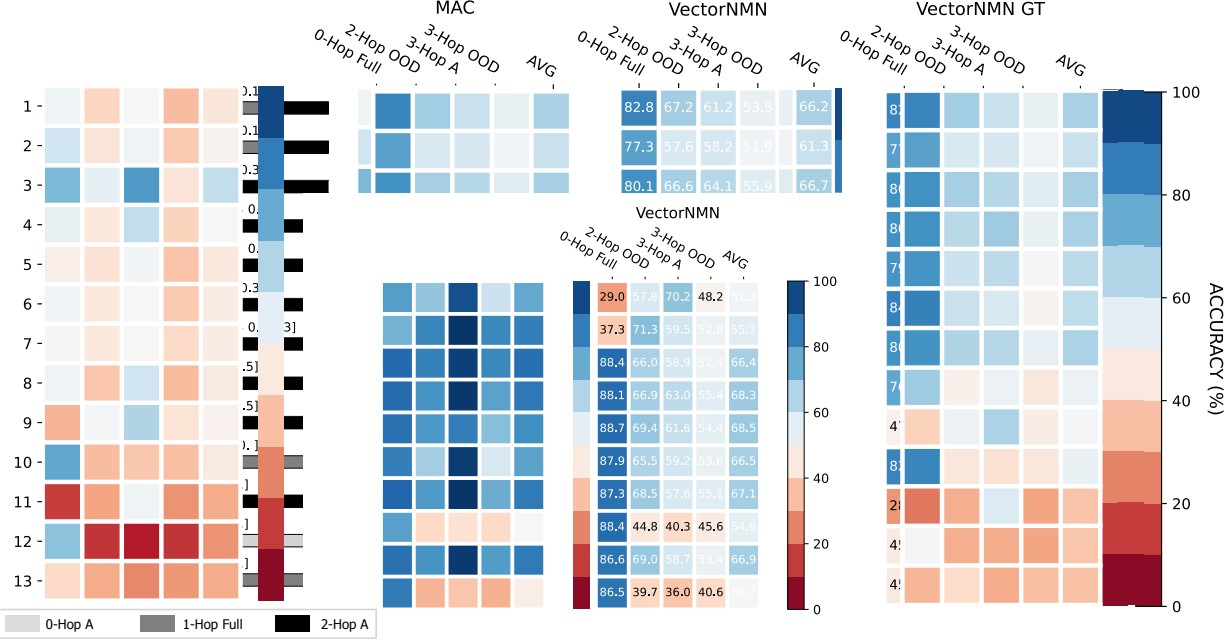

Figure 6: Training complexity distribution effect. The number of questions is kept fixed and is equal to 100k. Each row corresponds to a different training set where different fractions of questions from `0-Hop A`, `1-Hop Full`, and `2-Hop A` are selected. For each model, the accuracies on the respective test set are shown in each column. The average across the four test set is also shown in the `AVG` column.

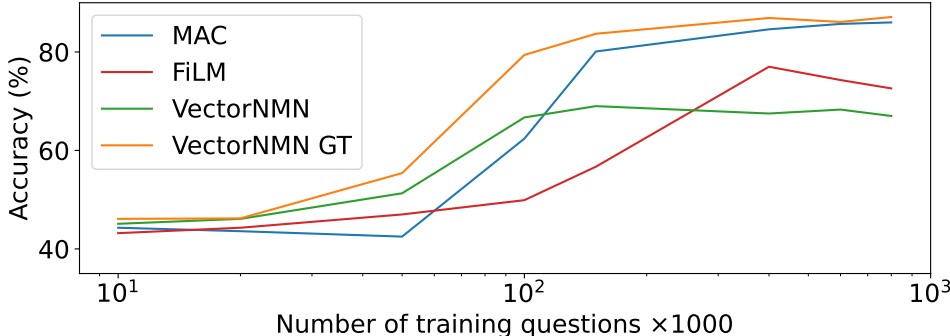

Figure 7: Average accuracy of the four test sets by varying the number of questions using fractions from dataset 3 in Figure 6.

The results obtained from the experiments conducted with different fractions of diversity are presented in Figure 6 for 100k questions. To provide a summary of the findings, we calculate the average result across the four test sets for each model and display it in the `AVG` column.

**VectorNMN has superior performance and is more robust to distribution changes in low data regimes.** We observe that when limited number of training questions are available, MAC performs poorly except for dataset number 3 and shows high sensitivity to distribution changes. VectorNMN on the other hand has more consistent results and achieves an overall better performance. The results can be further improved when program generator (VectorNMN GT) is provided.

**MAC gains strong systematic generalization performance and robustness to distribution changes when sufficient data is available.** To assess the impact of question complexity distribution in large data regimes, we identified the best set of fractions (dataset 3 in Figure 6) and progressively increased the number of questions until reaching saturation accuracy, as depicted in Figure 7. We determined that employing 600k questions would suffice for this dataset to observe the behavior of networks when using a large amount of data. Compared to low data case, the patterns alter when expanding the number of questions to 600k, as depicted in Figure 8. MAC becomes more robust to distribution changes as the amount of data increases and outperforms VectorNMN.

**Program generator plays an important role in systematic generalization.** The gap between VectorNMN and VectorNMN GT also increases as the number of training questions grows, indicating that the performance bottleneck of modular networks lies in learning the program generator, specifically in large data scenarios.

**Using abundant simple questions can lead to strong performance.** We also note that the inclusion of a distribution with more simple questions proves to be valuable in both data regimes, as it results in improved systematic generalization performance. Dataset 1 in Figure 6 and dataset 7 in Figure 8 show that it is not necessary to sample a large amount of complex questions to achieve high systematic generalization performance.

**Uniform sampling from different complexities can be sub-optimal.** MAC's performance on the uniform dataset (7th in Figure 6 and 6th in Figure 8) is relatively lower compared to the best results achieved. In contrast, modular networks show similar performance to other full distributions when using the uniform distribution.

## 4  Discussion

**Summary.**  We introduced Data Diversity Design (D3) for designing datasets with improved systematic generalization performance in VQA. We showed that diverse simple tasks significantly enhanced systematic generalization in VQA. We demonstrated that the inclusion of diverse questions, particularly those aligned with the particular systematic generalization aspects, led to better overall performance. Additionally, we investigated the impact of different sampling strategies from different question complexities and noted that distributions with enough simple and diverse questions can attain better systematic generalization results

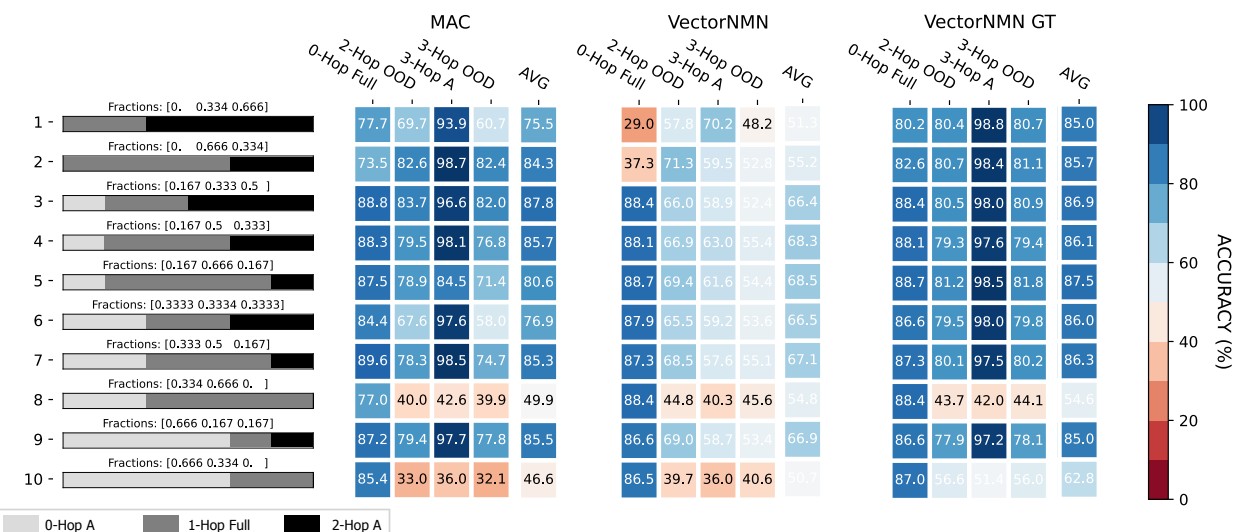

Figure 8: Training complexity distribution effect for 600k questions. Refer to Figure 6 for more details.

than the common uniform sampling practice. Our results showed that modular networks such as VectorNMN are more data efficient and more robust to question complexity distribution variations. On the other hand, the monolithic networks attain their best performance and robustness to distribution changes when sufficient data is available.

**Limitations.** While large language models Brown et al. (2020); Chowdhery et al. (2022); Taylor et al. (2022) have shown promise in addressing systematic generalization Anil et al. (2022); Wei et al. (2022a;b), their training data biases are not well-understood. This motivated us to focus on datasets created in controlled settings to isolate and investigate specific factors related to data diversity and systematic generalization. However, our studies may not fully capture the complexity and variability of real-world data.

While we have observed that diversity contributes to improved systematic generalization, further investigation is needed to fully understand the underlying reasons behind this effect. Our hypothesis is that diversity promotes modularity in neural networks, which in turn enhances systematic generalization. To provide additional insights into the modularity of networks, we present an experiment in Appendix C. Lastly, the evaluation is primarily based on the VQA task, and it would be valuable to investigate the transferability of the observed effects to other domains and tasks.

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

# Appendix

## A   Implementation Details

We used MAC Hudson & Manning (2018), FiLM Perez et al. (2018), Vectorized Neural Module Networks Andreas et al. (2016); Bahdanau et al. (2019a) (VectorNMN), its counterpart with given program generator (VectorNMN GT), and the version introduced in D'Amario et al. (2021) with modular image encoders (VectorNMNSepStem). To save time and computational resources, we excluded FiLM and VectorNMNSepStem from our experiments about question complexity distributions due to their inferior performance compared to MAC and VectorNMN. We conducted hyperparameter search using the attribute comparison and composition datasets and identified a set of effective hyperparameters that yielded consistent results. We searched for batch size, learning rate, number of iterations, and the way we do D3 for these datasets. We found that with 0.0001 as learning rate and 64 as batch size we get consistent results on the datasets. We tried training with early stopping but found that long training works better. We used 1M iterations for MAC and 500K iterations for the other methods. We also conducted experiments with curriculum learning in which we first training on simple questions from the D3 sets and then train with the other questions but found that training on a single mixed dataset achieves better results. Finally, we tried MAC and FiLM with ground-truth program as input instead of question. However, they resulted in poor performance for systematic generalization.

For program generator, we first tried the sequence-to-sequence model with attention of CLOSURE Bahdanau et al. (2019a) and found that it has poor performance for systematic generalization tasks. Every module in the sequence-to-sequence model is considered as separate instruction. For instance, `filter_color[gray]` is a separate instruction in the model. We observed that it is hard for the model to discriminate different modules when there is compositional biases in the dataset. So, inspired by Madan et al. (2022), we decided to separate the high level modules and their parameters in a two stream sequence-to-sequence model. In the two stream sequence-to-sequence, one stream provides the high level instruction, e.g., `filter` in the above example, and the other stream provide its parameter, e.g., `gray`. This change resulted in more than 40% increase in predicting the correct program for both datasets. All of our results for VectorNMN are based on the two-stream program generator.

## B   Effect of D3 Proportion

Figure 9 shows the effect of amount of proportions in D3 of `1-Hop Full` with `2-Hop A` and the results on `2-Hop OOD` for the VectorNMN GT model. We also compare D3 with 90% and 30% in Table 6 on the `2-Hop` dataset. We observe that for VectorNMNSepStem we have the best result with 90% simple questions. MAC and VectorNMN perform better when D3 with 30% is used. Please refer to Section 3.3 for a more granular effect of different proportions on the final results.

Table 6: *Accuracy of D3 with* 30% *proportion compared with* 90% *on* `2-Hop OOD` *(top) and* `3-Hop OOD` *(bottom).*

| Training dataset | FiLM | MAC | VectorNMN | VectorNMNSepStem |
|---|---|---|---|---|
| 2-Hop A | 31.3 | 29.2 | 36.7 | 36.1 |
| + D3 (`1-Hop FULL (30%)`) | 46.9 | 64.0 | 61.0 | 54.8 |
| + D3 (`1-Hop FULL (90%)`) | 50.0 | 55.5 | 50.9 | 67.2 |
| 2-Hop A | 28.5 | 27.2 | 35.9 | 36.0 |
| + D3 (`1-Hop FULL (30%)`) | 44.1 | 54.4 | 51.8 | 50.7 |
| + D3 (`1-Hop FULL (90%)`) | 48.2 | 46.5 | 49.3 | 56.2 |

## C   Testing Modularity

To test the modularity of the trained networks, we devised the following experiments. We chose the `0-Hop` with a single attribute as the test set. We can use this test as a proxy for measuring the degree of modularity

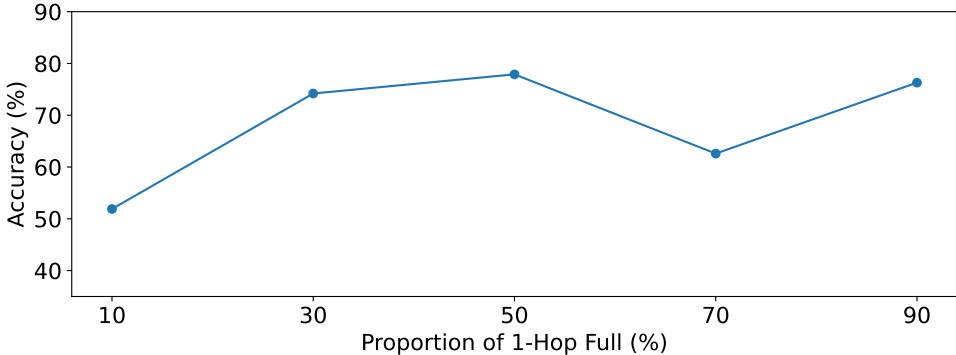

Figure 9: Effect of D3 with `1-Hop Full` proportions on accuracy. The base training set is `2-Hop A` and the test set is `2-Hop OOD`.

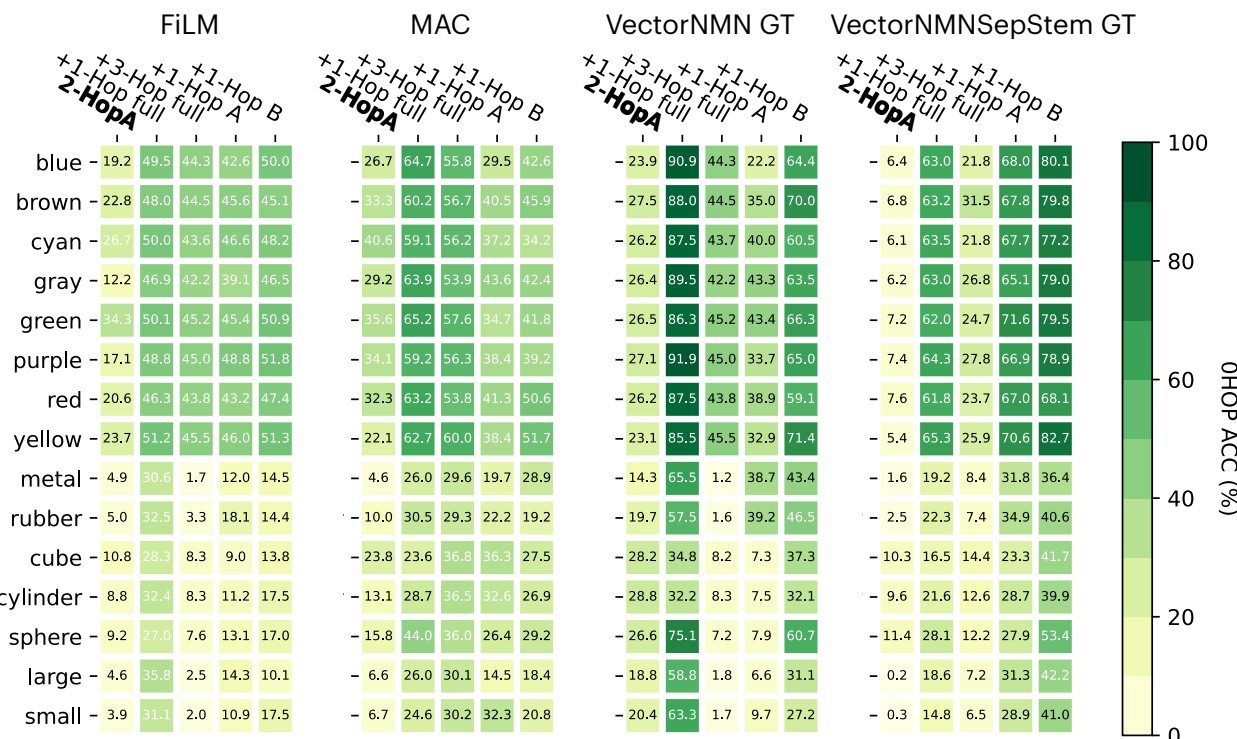

Figure 10: Effect of applying D3 on modularity. Modularity is measured by accuracy on the questions with a single attribute from the `0-Hop` test set.

since this set contains questions about one attribute of the objects. Figure 10 shows the results of different training regimes defined for the `2-Hop` dataset for each attribute. Interestingly, D3 with `1-Hop Full` results in better modularity. The color attributes are better learned than the other attributes. There is also a notable accuracy difference between `Sphere` and other shapes. The reason for this difference is the biases in the images between CoGenT A and CoGenT B, where `Cube` and `Cylinder` objects appear with different colors. We also note that even though VectorNMN GT is using the ground-truth program generator, training `2-Hop A` results in poor modularity, showing that modules are not performing their expected task when there are biases in the dataset. Additionally, we note that D3 with `1-Hop A` usually improves modularity, although it only adds length diversity.

Table 7: *Accuracy of base training on* `2-Hop A`.

| Model | 0-Hop | 2-Hop OOD | 3-Hop A | 3-Hop OOD |
|---|---|---|---|---|
| VectorNMNSepStem | 20.8 | 36.1 | 60.4 | 36.0 |
| VectorNMN | 22.3 | 36.7 | 57.9 | 35.9 |
| MAC | 20.9 | 29.2 | 55.8 | 27.2 |
| FiLM | 13.9 | 31.3 | 53.8 | 28.5 |
| MiniGPT-v2 | 60.5 | 70.9 | 73.7 | 67.6 |

## D    Question Generation

The question generation code is based on the original CLEVR Johnson et al. (2017a)[3] and CLEVR-IEP Johnson et al. (2017b)[4] dataset generation frameworks. We added our constraints to the code base to generate the biased questions. We have made our question generation code for the CLEVR dataset publicly available[5].

## E    Base Results of 2-Hop A

Accuracy of base training on `2-Hop A` dataset for different models is shown in Table 7. The change in accuracy results shown in the heatmaps of Figure 3 and Figure 5 are relative to Table 7.

## F    Symbolic Methods

**NS-VQA Yi et al. (2018).**    The original NS-VQA framework comprises three components: 1) a scene parser, which segments objects in the input image and reconstructs a structural scene representation; 2) a question parser, responsible for converting a question into a program; and 3) a program executor that executes the program on the structural scene representation to derive an answer.

To circumvent the need for additional annotation of segmentation masks in our generated datasets and to mitigate biases inherent in object segmentation models, we utilize the original scene graphs used in scene generation. The role of the question parser aligns with the program generator in VectorNMN, which we have already employed. Subsequently, the program executor can effortlessly execute the generated program on the provided scene graph to produce an answer. Our implementation of NS-VQA can be considered an upper bound compared to the original NS-VQA, as it employs ground-truth scene graphs instead of conducting object segmentation. Therefore, the only aspect that can be learned is the program generator, which we have already evaluated for VectorNMN and discussed in Appendix A. Nevertheless, for completeness, we adhered to the exact implementation of the sequence-to-sequence model used in NS-VQA as the question parser in our NS-VQA and evaluated the complete results on the `2-Hop` datasets. The results of NS-VQA are provided in Table 8. Similar to the previous results, D3 enhances our implementation of NS-VQA in most of the cases.

Table 8: *Results of NS-VQA on* `2-Hop` *datasets*.

| Dataset | 0-Hop | 2-Hop OOD | 3-Hop A | 3-Hop OOD |
|---|---|---|---|---|
| `2-Hop A` | 10.77 | 16.02 | 37.71 | 12.12 |
| `+1-Hop A` | 11.76 | 20.97 | 43.28 | 17.48 |
| `+1-Hop B` | 35.01 | 15.88 | 43.61 | 12.54 |
| `+1-Hop FULL` | 18.12 | 31.70 | 44.25 | 19.38 |
| `+3-Hop FULL` | 10.00 | 99.38 | | |

---

[3]https://github.com/facebookresearch/clevr-dataset-gen
[4]https://github.com/facebookresearch/clevr-iep
[5]https://github.com/AmirooR/D3QuestionGenerationCLEVR

**ViperGPT Surís et al. (2023).** Viper-GPT relies heavily on pretrained models and does not undergo specific training. Therefore, systematic generalization testing cannot be conducted on Viper-GPT. As pointed out by the authors of Viper-GPT: "*ViperGPT may inherit biases from the pretrained models it uses. These biases may be reflected in the outputs generated by our model. It is recommended to consider this potential bias when using ViperGPT and interpreting its outputs.*" We also note that Modular Networks are similar to symbolic networks with having their modules learned through training.

