# OpenReview forum: "D3: Data Diversity Design for Systematic Generalization in Visual Question Answering"
_TMLR — Accepted by TMLR_

### Review · Reviewer_bvDM · 2024-01-27

**Summary Of Contributions:**

The paper discusses Visual question-answering tasks and revolves around the effect of different compositional skills in the training data. Specifically, the paper separate skills to the 0-Hop set, which evaluates generalization to shorter, single attribute questions; the 2-Hop OOD set, which assesses attribute composition generalization in more complex questions; the 3-Hop A set, which tests the ability to generalize to longer questions; and the 3-Hop OOD set, evaluating of both out-of-distribution length and attribute composition generalization.
The results of the paper show that distributing the training set differently can significantly affect the performance of the same architecture, which is an important conclusion given that most of the literature mostly revolves around architecture and not the training data.

**Audience:**

Yes

**Broader Impact Concerns:**

The Broader Impact Concerns were provided, suggesting that the authors did not use real images in the study due to certain concerns. In my opinion, these concerns are not sufficient reasons to avoid using real data.

**Claims And Evidence:**

No

**Requested Changes:**

* Consider adding a section on real images and non-synthetic questions.
* It would be helpful if you could discuss symbolic solutions.
* Including qualitative examples would enrich your paper.
* Incorporating more experiments from the appendix and improving organization could enhance the paper's clarity.

**Strengths And Weaknesses:**

Strengths:
(+) The paper is unique in that it focuses on the data, not the architecture. It attempts to separate training into different skills and demonstrates that the distribution of these skills in the training data can indeed impact the ability to generalize.
(+) The presentation and articulation of the results were done well; it was very easy to read through, grasp the main contributions, and understand the conclusions.

Weaknesses:
(W1) The work should be expanded. At the moment, it provides good proof that there is a need to sample training data better. However, for it to be significant, such a study needs also to be conducted on real image data, not just syntactic. Datasets of VQAv2, TallyQA, and others could provide more significant insights. Ideally, this kind of study should reveal biases in any VQA setup, and it does require more work to understand how nonsynthetic questions can be broken down into different relational-compositional skills. An attempt to incorporate GQA in the appendix, which generates questions syntactically on real images, but it's quite limited compared to the study on CLEVR, and in any case, should not be present only in the appendix.
(W2) I would also suggest a section for symbolic solutions, such as Neural-Symbolic VQA: Disentangling Reasoning from Vision and Language Understanding, and up to the recent study of ViperGPT: Visual Inference via Python Execution for Reasoning.

---

> ### Author Response · Authors · 2024-04-25
> **Response to bvDM**
>
> Thank you for your thoughtful review and the emphasis on expanding our work into real-world datasets.
>
> We agree that biases in training data are a critical concern and appreciate the literature that endeavors to identify and mitigate these biases. We would like to clarify that our study's objective was not to discover biases in VQA datasets but to empirically show how diversity in controlled training environments influences systematic generalization. This focus allows us to distill the direct effects of task diversity without the convolution of uncontrolled biases, which is often a challenge in real-world datasets.
>
> In response to your feedback, we have included additional results using real-image datasets and non-synthetic questions, such as TallyQA and GQA, to provide a more comprehensive understanding of our findings’ applicability. Furthermore, our inclusion of the MiniGPT-v2 architecture and NS-VQA (see below) expands the scope and relevance of our study.
>
> We hope our additional experiments and discussions in the revised manuscript affirm the value of our contributions to the understanding of systematic generalization and data diversity across various VQA models and settings. Our findings lay the groundwork for future inquiries into bias identification and mitigation within real-world data.
>
> **Symbolic Methods:**
> - **NS-VQA:** The original NS-VQA framework comprises three components: 1) a scene parser, which segments objects in the input image and reconstructs a structural scene representation; 2) a question parser, responsible for converting a question into a program; and 3) a program executor that executes the program on the structural scene representation to derive an answer.
>
> To circumvent the need for additional annotation of segmentation masks in our generated datasets and to mitigate biases inherent in object segmentation models, we utilize the original scene graphs used in scene generation. The role of the question parser aligns with the program generator in VectorNMN, which we have already employed. Subsequently, the program executor can effortlessly execute the generated program on the provided scene graph to produce an answer. Our implementation of NS-VQA can be considered an upper bound compared to the original NS-VQA, as it employs ground-truth scene graphs instead of conducting object segmentation. Therefore, the only aspect that can be learned is the program generator, which we have already evaluated for VectorNMN and discussed in Appendix A. Nevertheless, for completeness, we adhered to the exact implementation of the sequence-to-sequence model used in NS-VQA as the question parser in our NS-VQA and evaluated the complete results on the 2-Hop datasets. The results of NS-VQA are provided below (Table 9 in the revised paper).
>
> |   Train set  |   0-HOP  |   2-HOP OOD  |   3-HOP OOD  |   3-HOP A  |
> |---|---|---|---|---|
> |   2-HOP A  |   10.77  |   16.02  |   12.12  |   37.71  |
> |   +1-HOP A  |   11.76  |   20.97  |   17.48  |   43.28  |
> |   +1-HOP B  |   35.01  |   15.88  |   12.54  |   43.61  |
> |   +1-HOP Full  |   18.12  |   31.70  |   19.38  |   44.25  |
> |   +3-HOP Full  |   10.0  |   99.38  |     |     |
>
> Similar to the previous results, D3 enhances our implementation of NS-VQA in most of the cases.
>
> - **ViperGPT:** Viper-GPT relies heavily on pretrained models and does not undergo
> specific training. Therefore, systematic generalization testing cannot be conducted on Viper-GPT. As pointed
> out by the authors of Viper-GPT: “***ViperGPT may inherit biases from the pretrained models it uses. These
> biases may be reflected in the outputs generated by our model. It is recommended to consider this potential
> bias when using ViperGPT and interpreting its outputs.***” We also note that Modular Networks are similar to
> symbolic networks with having their modules learned through training.
>
> We have added a section on symbolic methods in Appendix F.

---

> > ### Comment · Reviewer_bvDM · 2024-05-24
> >
> > Thank you for the comments and the additional experiments. I believe the paper now presents sufficient evidence, and thus, I am inclined to recommend acceptance. However, I would suggest that the authors reorganize the paper to better reflect its robustness across different types of VQA datasets.

---

### Review · Reviewer_8yzK · 2024-02-19

**Summary Of Contributions:**

This paper provides new evidence to understand how different diversity affects systematic generalization in VQA, which reveals that the diversity of simple tasks plays an important role in achieving systematic generalization. Moreover, the paper also provides some other findings to understand the interactions between data diversity design, neural network architectures, and systematic generalization capabilities.

**Audience:**

Yes

**Broader Impact Concerns:**

None.

**Claims And Evidence:**

Yes

**Requested Changes:**

1. The visualization analysis or other quantitative analysis of the experimental results.
2. More datasets.
3. Comparison with large models like MiniGPT4 v2.

**Strengths And Weaknesses:**

Strength

1. The paper is interesting and makes sense.
2. The research question is interesting, and the conclusions also seem to align well with common sense.
3. The experimental setup is well done.

Weakness

1. As mentioned in the conclusion of the article, the experiment did not provide an analysis of why diverse simple tasks yield better results. It would be beneficial if visualization or other quantitative analyses were presented.
2. Hope to see more experiments beyond just CLEVR. It would be even better to have more datasets, attributes, and architectures available.
3. Can comparisons or adjustments be made using large models? For example, MiniGPT4 v2.

---

> ### Author Response · Authors · 2024-04-25
> **Response to 8yzK**
>
> Thank you for your constructive feedback and for emphasizing the importance of providing a deeper analysis into why diversity of simple tasks leads to better systematic generalization in VQA models.
>
> Our primary goal in this study was to empirically demonstrate the impact of data diversity on systematic generalization across different tasks, datasets, and network architectures. We appreciate your suggestion to further investigate the underlying reasons why this diversity leads to improved performance. While our hypothesis is that networks trained on diverse simple tasks become more modular, thus enhancing their ability to learn and effectively combine subtasks, we did not initially set out to probe the mechanisms behind this empirically observed phenomenon in depth.
>
> We have included additional visualizations and quantitative analyses in the revised manuscript, which illustrate the performance improvements across a range of conditions pertinent to task diversity. Furthermore, we provided empirical substantiation for our hypothesis in appendix C by presenting results indicating increased modularity due to training with diverse simple tasks.
>
> Although our hypothesis posits that modularity may be central to this improved performance, we wish to clarify that our study's main contribution is to establish the empirical relationship between diversity and generalization. Investigating the precise mechanisms by which modularity—or other potential factors—contributes to this performance is a complex question that could form the basis of future research. Nevertheless, we believe that the materials we have provided will help to substantiate our claims regarding diversity and offer a starting point for subsequent inquiries into why this relationship exists.
>
> Please refer to the "General Response" we provided above for experiments using additional datasets and architectures including MiniGPTv2.
>
> Thank you for your valuable feedback, which has helped us refine our presentation and strengthen the paper.

---

> > ### Comment · Reviewer_8yzK · 2024-05-23
> >
> > I've read the author response and the modified version of the paper. I see that all my concerns have been addressed. I have no more questions.

---

### Review · Reviewer_onP2 · 2024-04-02

**Summary Of Contributions:**

The paper investigate the role of data, and model architecture for VQA generalization.

**Audience:**

Yes

**Broader Impact Concerns:**

No concerns.

**Claims And Evidence:**

No

**Requested Changes:**

Explain why the problem (dataset diversity on generalization for VQA) is worth study? Can it be broke down into subtasks. What is is the impact of this study, and why the findings are useful?

**Strengths And Weaknesses:**

Strength:
- created a data diversity benchmark - using the proposed D3 (Data Diversity Design)
- explores model's generalization using attributes composition task
- tracks the question complexity distribution and length
- the experiment results show the effect of dataset diversity on generalization


Weakness:
- the paper's design choices and results are scattered and very hard to follow
- the paper lacks an overview of context. I don't follow why the proposed design and results are important

---

> ### Author Response · Authors · 2024-04-25
> **Response to onP2**
>
> Thank you for your review.
>
> We re-iterate the strengths of the paper pointed by other reviews here for further clarification:
> - Reviewer sviB mentioned that "*The problem that our work tackles is an important problem – what specific aspects of data diversity can improve the systematic generalization in the VQA task. This work curates a new diversity benchmark on VQA. and experimental results reveal several findings as highlighted with bold fonts in section 3.*"
> - Reviewer 8yzK pointed out that "*The paper is interesting and makes sense. The research question is interesting, and the conclusions also seem to align well with common sense. The experimental setup is well done.*"
> - Reviewer bvDM suggested that "*The paper is unique in that it focuses on the data, not the architecture. It attempts to separate training into different skills and demonstrates that the distribution of these skills in the training data can indeed impact the ability to generalize. The presentation and articulation of the results were done well; it was very easy to read through, grasp the main contributions, and understand the conclusions.*"
>
> Our work delves into the crucial realm of systematic generalization in VQA tasks, aligning with the broader context of understanding human intelligence and its ability to apply knowledge gained from related subtasks to solve novel challenges, as highlighted by Lake et al. (2019). Recent studies, such as the work by [2], have underscored the importance of compositional skills in large language models (LLMs) and their emergent behavior.
>
> Deep learning methods have historically grappled with achieving systematic generalization, particularly in the face of complex compositional and structural biases present in training data. While datasets inherently carry biases [3], recent research [4], show the lack of robustness in real-world scenarios for large vision language models like BLIP [5]. Our study fills a crucial gap in understanding the specific aspects of data diversity that contribute to improved systematic generalization and investigates the conditions under which these factors can be effectively leveraged. In line with recent findings (Sorscher et al., 2022), we challenge the notion of solely relying on neural scaling laws (Gordon et al., 2021; Hestness et al., 2017; Hoffmann et al., 2022; Rosenfeld et al., 2020; Zhai et al., 2022) to enhance model performance through data scaling. Instead, we advocate for strategic dataset design based on factors such as difficulty and dataset size, which can lead to greater cost efficiency without compromising performance.
>
> We believe our research addresses a pressing need in the field and contributes to advancing the understanding and practical application of data diversity in enhancing systematic generalization for VQA tasks.
>
> References:
>
> [2] Arora, S., & Goyal, A. (2023). A Theory for Emergence of Complex Skills in Language Models.
> [3] Kim, Y., et al. (2024). Discovering and Mitigating Visual Biases through Keyword Explanation.
> [4] Park, S., et al. (2023). RoCOCO: Robustness Benchmark of MS-COCO to Stress-test Image-Text Matching Models.
> [5] Li, J., et al. (2022). Blip: Bootstrapping language-image pretraining for unified vision-language understanding and generation.

---

### Review · Reviewer_sviB · 2024-04-25

**Summary Of Contributions:**

This work studies the relations between data diversity and models’ systematic generalization in the VQA task. It considers two aspects of data diversity: attribute composition and reasoning hops. It performs experiments using multiple VQA models, including monolithic architectures and neural module networks, on a newly tailored data diversity benchmark based on the CLEVR repository.

**Audience:**

Yes

**Broader Impact Concerns:**

No specific ethical concern in this work.

**Claims And Evidence:**

Yes

**Requested Changes:**

Please refer to the weakness above.

**Strengths And Weaknesses:**

<Strengths>
1. The problem that this work tackles is an important problem – what specific aspects of data diversity can improve the systematic generalization in the VQA task.

2. This work curates a new diversity benchmark on VQA.

3. Experimental results reveal several findings as highlighted with bold fonts in section 3.

<Weaknesses>
1. Given that this work can be regarded as an empirical study, the performed evaluations are highly limited.
- Only a single dataset is used – CLEVR.
- Only two attributes, four different hops, and four question types are considered for the test set.
- Although it could be a meaningful initial step, it is highly limited to draw dependable conclusions. Moreover, its scope of analyses is relatively narrow as a journal submission.

2. The tested models are outdated.
- Five models are tested but they are proposed prior to 2020.
- Thus, the absolute accuracy scores reported in Table 1 are quite low.
- This work mostly considers module networks based VQA models. Why do not test other categories of VQA model?
- It would be more interesting to see what results can be obtained on recent powerful LLMs.

3. No qualitative results are presented for the VQA tasks.

---

> ### Author Response · Authors · 2024-04-25
> **Response to sviB**
>
> Thank you for your thorough evaluation of our work and for acknowledging the significance of our research on data diversity's impact on systematic generalization in VQA tasks, as well as our efforts in curating a new diversity benchmark in this domain. We appreciate your insights and have considered your feedback carefully to enhance the quality of our paper.
>
> Regarding the weaknesses you highlighted:
>
> - **Limited Experimental Scope:** We understand your concern about the evaluations conducted within a limited scope, and we agree that expanding our experiments can strengthen our findings. To address this, we have augmented our experiments to include additional datasets beyond CLEVR, such as the GQA dataset and TallyQA, which cater to a broader range of attributes and question types. This expansion allows us to validate our claims on a more diverse collection of data, and these new results have been included in the revised manuscript (refer to Section 3.2 and Tables 2,3,4).
>
> - **Diversity of Models:** You raised an important point regarding the models tested in our study. We have now included additional experiments involving more recent and advanced models. Our extended analysis now encompasses recent large language models such as GPT-2, and Llama-2 (in MiniGPTv2) to evaluate their performance on the VQA task with diverse data. This provides a more comprehensive overview of the landscape of VQA models and how each benefits from data diversity. The updated comparison, including these contemporary models, can be found in Tables 4,5,6 of the revised manuscript. We do not see any base for considering the results of Table 1 to be low without having comparisons. As mentioned in the general response about MiniGPTv2, their off-the-shelf trained model has a poor performance on our 2-Hop OOD test set although it has been trained extensively on large VQA datasets and their primary goal is to have a model that excels in multiple tasks.
>
> - **Qualitative Results:** In response to the absence of qualitative results in our initial submission, we have now incorporated qualitative examples on our 2-Hop OOD test set in Section 3.1 and Figure 4.
>
> We believe that the additions we have made thoroughly address your concerns and serve to strengthen our results and conclusions.
>
> We are grateful for your valuable feedback and hope that our revisions will meet the journal's standards for empirical research in this field.

---

### Decision · Action_Editor_qzBC · 2024-06-03

**Recommendation:** Accept as is

**Comment:**

The paper provides a systematic study to investigate VQA generalization. Coupled with real-data experiments, the experimental evidence is sufficiently convincing. The topic of dataset design is relevant to current VQA research.

**Audience:**

Dataset construction has become a major focus for VQA (and vision/language in general) research, therefore I believe that this work is timely, and would be of interest to the community. The reviewers highlighted that the absence of non-synthetic data in the first draft may limit interest, however, the reviewers agree that the final version addresses this.

One of the main conclusions is that, sufficiently diverse, simple (requiring few reasoning steps) training tasks are a driver of good generalization. This is an encouraging finding, since it may indicate that, with appropriate data collection, one may not need vast amounts of data to cover more complex questions.

**Claims And Evidence:**

The paper presents a study on dataset design for VQA, in particular focussing on systematic generalization to novel compositions of attributes and novel composition chain lengths. The paper proposes D3, a strategy for creating a training set that elicits improves "compositional" and "length" generalization. The main focus of the study is on CLEVR, a synthetic environment without confounders, where these factors can be studied systematically. The paper also includes TallyQA and GQA non-synthetic datasets that supports the finding that D3 improves performance. Testing on a few different model classes, (VectorNMN, MAC, LLM-based), the paper provides sufficient evidence to support their insights and strategy for dataset composition.